# Structure of the *Neisseria meningitidis* Type IV pilus

Subramania Kolappan[1], Mathieu Coureuil[2], Xiong Yu[3], Xavier Nassif[2], Edward H. Egelman[3] & Lisa Craig[1]

*Neisseria meningitidis* use Type IV pili (T4P) to adhere to endothelial cells and breach the blood brain barrier, causing cause fatal meningitis. T4P are multifunctional polymers of the major pilin protein, which share a conserved hydrophobic N terminus that is a curved extended α-helix, α1, in X-ray crystal structures. Here we report a 1.44 Å crystal structure of the *N. meningitidis* major pilin PilE and a ∼6 Å cryo-electron microscopy reconstruction of the intact pilus, from which we built an atomic model for the filament. This structure reveals the molecular arrangement of the N-terminal α-helices in the filament core, including a melted central portion of α1 and a bridge of electron density consistent with a predicted salt bridge necessary for pilus assembly. This structure has important implications for understanding pilus biology.

---

[1] Department of Molecular Biology and Biochemistry, Simon Fraser University, Burnaby, British Columbia, Canada V5A 1S6. [2] Institut Necker-Enfants Malades, INSERM U1151, Université Paris Descartes, 14 Rue Maria Helena Vieira Da Silva, CS 61431, 75014 Paris, France. [3] Department of Biochemistry and Molecular Genetics, University of Virginia School of Medicine, Charlottesville, Virginia 22908, USA. Correspondence and requests for materials should be addressed to E.H.E. (email: egelman@virginia.edu) or to L.C. (email: licraig@sfu.ca).

Neisseria meningitidis (Nm) is one of the very few extracellular bacterial pathogens able to cross the blood brain barrier after having invaded the bloodstream from the nasopharynx. In the bloodstream an unusually tight interaction between the bacteria and the brain microvasculature endothelial cells is established, leading to cortical plaque formation that results in opening of the blood brain barrier and bacterial invasion of the brain[1–5]. This interaction is mediated by Type IV pili (T4P), long thin polymers of pilin proteins that interact with two endothelial cell receptors, CD147 and the β2 adrenergic receptor (β2AR). The interaction with CD147 is responsible for bacterial adhesion[6] and the interaction with the β2AR induces signalling in endothelial cells[7]. While the host cell signalling events that lead to disruption of the blood brain barrier are somewhat well-defined, the mechanism by which T4P initiate this process is poorly understood and would benefit from a detailed structure of the N. meningitidis T4P.

X-ray crystal structures of full-length Type IV pilin proteins have been obtained by dissociating intact pilus filaments with detergent, revealing a canonical structure with an extended gently curving 53-amino-acid α-helix (α1), the C-terminal half of which is embedded in a globular C-terminal domain[8–12]. The N-terminal half of the α-helix, α1N, protrudes from the globular domain and is comprised of hydrophobic residues, with the exception of a threonine or serine at position 2 and an invariant glutamate at position 5. Two helix-breaking residues, Pro22 and Gly42, which are conserved in Type IV pilins of the IVa class, introduce kinks in α1 and are responsible for its curvature. An additional glycine at position 14 is also conserved. α1N has dual functions in T4P biogenesis: it anchors the globular domain in the inner membrane before pilus assembly[13], and it interacts with adjacent α1Ns in the assembled pilus, forming a staggered helical array in the filament core[8,9,14,15]. The conserved Glu5 is critical for T4P assembly[14,16–22] and models of T4P, based in part on the crystal structure of the full-length pilin subunit from Neisseria gonorrhoeae, PilE (Ng PilE), place this side chain in a salt bridge with the positively charged N-terminal amine of its neighbouring molecule in the otherwise hydrophobic core of the filament[8,9,11,14,23].

N. meningitidis PilE (Nm PilE) is 78% identical to Ng PilE and 100% identical in α1 (Fig. 1a,b). Both pilins share a ~20 amino-acid hypervariable region, located between two conserved cysteines, imparting antigenic variability to these pili[24–26]. In N. gonorrhoeae PilE the hypervariable region is located on a β-hairpin that lies on top of the globular domain β-sheet[9,11]. A cryo-electron microscopy (cryoEM) reconstruction of the Ng T4P was determined at 12.5 Å resolution and an atomic model was built by fitting the full-length Ng PilE structure into the cryoEM density[9]. The PilE globular domains fit well into the reconstruction, exposing the hypervariable region prominently on the pilus surface and burying the N-terminal α-helices in the filament core. PilE subunits are related in the Ng T4P reconstruction by an axial rise of 10.5 Å and an azimuthal rotation (twist) of 100.8°. The subunits are held together largely by hydrophobic interactions among the N-terminal α-helices but these helices were not resolved due to the limited resolution cryoEM map. The importance of the hydrophobic interactions was corroborated by biochemical studies, which showed that Ng T4P require detergent to dissociate into pilin subunits[9,11,14]. The interactions between subunits are very strong, requiring temperatures of 60 °C to denature the filaments[14], consistent with their ability to withstand tensile forces of 100 pN or more[27–31]. Yet both Ng and Nm T4P have been shown to undergo a force-induced conformational change that exposes an epitope, EYYLN, along the length of the pilus[29,32]. EYYLN is located at the end of α1 (resides 49–53) in Ng PilE, a site that is buried by subunit:subunit interactions in the Ng T4P model and only accessible at the tips of the pili[9]. Consistent with this model, anti-EYYLN antibodies normally bind to the tips of Nm and Ng T4P but not along its length[33,34]. Yet when Ng T4P are placed under force using optical or magnetic tweezers or molecular combing they become longer and narrower[29]. This force-induced conformational change exposes the anti-EYYLN epitope all along the length of the pilus, allowing antibody to bind. Similarly, Brissac et al. showed that anti-EYYLN antibodies can bind along the length of Nm T4P when Nm are adhered to endothelial cells, a property that is dependent on expression of the minor pilin, PilX[32]. It is unclear how these T4P extend in a reversible manner without completely disrupting the extensive hydrophobic interactions among the N-terminal α-helices in the core of the filament. Another poorly understood phenomenon of T4P from at least one species, Geobacter sulfurreducens, is their ability to facilitate long-range electron transport, a property that is thought to occur via overlapping π–π orbitals of aromatic residues in α1 (refs 35,36). Higher resolution structures may provide a molecular basis for these and other remarkable properties of T4P.

Here we report the 1.44 Å X-ray crystal structure of Nm PilE and a cryoEM reconstruction of the Nm pilus at higher resolution, ~6 Å, than any T4P structure reported previously. Portions of the N-terminal α-helices are clearly resolved in this reconstruction, allowing their precise placement, which provides evidence of a salt bridge between Phe1:N and the Glu5 side chain, and reveals a melting of the central portion of α1N. This structure provides a molecular framework for understanding key aspects of T4P biology.

## Results

**Structure of the major pilin PilE from N. meningitidis.** N. meningitidis ΔN-PilE, comprising residues 29–161 of PilE from the high adhesive SB variant of N. meningitidis strain 8013 (ref. 37) was expressed with an N-terminal His-tag. This PilE variant, when expressed as a maltose-binding protein fusion on Staphylococcus aureus, induces recruitment of the β2AR in endothelial cells[38]. ΔN-PilE crystals were grown in the $P2_12_12_1$ space group and the structure was solved by molecular replacement with the full-length PilE structure from N. gonorrhoeae (Ng PilE, 2HI2[9]), which shares 78% sequence identity with N. meningitidis PilE (Nm PilE; Fig. 1a). Data collection and refinement statistics are reported in Table 1. Nm ΔN-PilE is a globular protein with an N-terminal α-helix corresponding to α1C, the C-terminal half of α1 (Fig. 2a), embedded in a four-stranded antiparallel β-sheet (β1 to β4, residues 77–121). Between α1C and the β-sheet is the αβ-loop (residues 54–76), an extended segment with a single-turn α-helix and a $3_{10}$ helix that forms one edge of the globular domain. Two residues in the αβ-loop, Ser63 and Ser69, are post-translationally modified in the native PilE[39,40]. On the opposite side of the β-sheet, the polypeptide chain exits β4 and forms a β-hairpin (β5–β6) that lies atop the β-sheet, followed by an irregular segment that protrudes from the globular domain surface then wraps under it, ending in an extended C terminus at the edge opposite the αβ-loop. Conserved disulfide-bonded cysteines at the end of β4 (Cys120) and close to the C terminus (Cys154) define the boundaries of the D-region containing the β-hairpin. The most protruding segment of the D-region, the β-hairpin loop and β6 plus the beginning of the loop that follows β6, is highly variable in sequence among N. meningitidis Type IV pilins and is referred to as the hypervariable region (residues ~127 to 146)[38]. The β-hairpin loop hooks over the top of the β-sheet and across

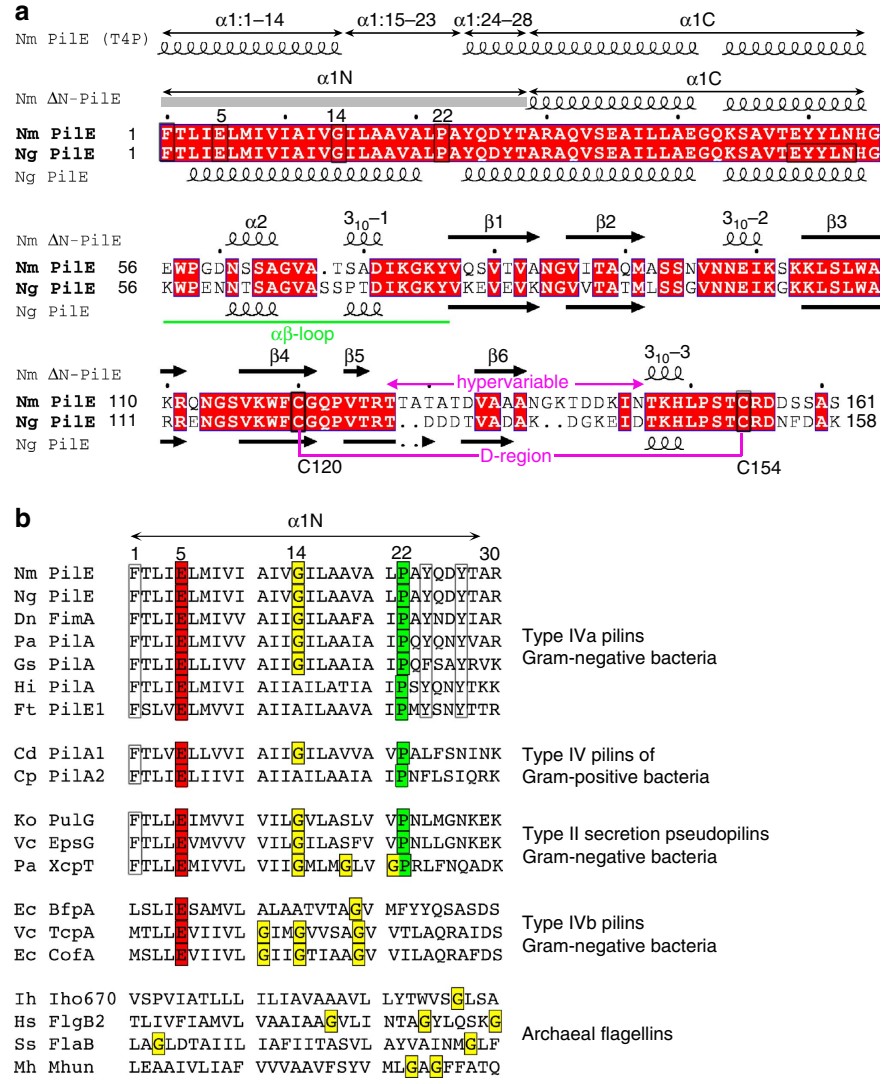

**Figure 1 | Sequence alignment between *N. meningitidis* and *N. gonorrhoeae* PilE and other Type IV pilin-like proteins. (a)** Amino-acid sequence alignment between *N. meningitidis* (Nm) and *N. gonorrhoeae* (Ng) PilE, with secondary structures indicated above and below the sequences, respectively, based on the crystal structures of Nm ΔN-PilE (residues 29–161, this study) and full-length Ng PilE[9,11]. A grey bar represents α1N, which is absent in the Nm ΔN-PilE structure. The secondary structure of Nm PilE α1 within the T4P filament is indicated at the top of the panel based on the cryoEM density map. The amino-acid alignment was performed using CLUSTAL OMEGA[69] with manual adjustments and visualized using ESPript[70]. Identical residues are shown in white font on a red background. Residues of interest are boxed and the conserved disulfide-bonded cysteines are connected. **(b)** N-terminal amino-acid sequence alignment among Type IV pilins from Gram negative and Gram positive bacteria, Type II secretion pseudopilins and archaeal flagellins. Conserved residues are boxed and/or highlighted. Glycines are also highlighted in the archaeal flagellins although their positions are not conserved. Nm PilE, *N. meningitidis* PilE (GenBank accession number WP_014573675), Ng PilE, *Neisseria gonorrhoeae* PilE (for 2HI2, P02974), Dn FimA, *Dichelobacter nodosus* (X52403), Pa PilA, *Pseudomonas aeruginosa* PilA (P02973); Gs, *Geobacter sulfurreducens* PilA (2M7G_A); Ft PilE1, *Francisella tularensis* Schu S4 PilE1 (CAG45522); Hi, non-typeable *Haemophilus influenzae* (AAX87353); Cd PilA1, *Clostridium difficile* PilA1 (ref. 71); Cp PilA2, *Clostridium perfringens* PilA2 (NP_563200); Ko PulG, *Klebsiella oxytoca* PulG (S11917); Vc EpsG, *V. cholerae* EpsG (AAA58788); Pa XcpT, *P. aeruginosa* XcpT (BAT65239); Ec BfpA, enteropathogenic *E. coli* BfpA (Z68186); Vc TcpA, *V. cholerae* TcpA (ABQ19609); Ec CofA, enterotoxigenic *E. coli* CofA (BAA07174); Ih Iho670, *Ignicoccus hospitalis* Iho670 (WP_011998704); Hs FlgB2, *Halobacterium salinarum* FlgB2 (CAP13655); Ss FlaB, *Sulfolobus sulfataricus* FlaB (AKA77968); Mh Mhun, *Methanospirillum hungatei* Mhun_3140 (ABD42825).

the β1–β2 loop; Ala129 at its apex forms a main chain hydrogen bond with Trp57 at the beginning of the αβ-loop.

Another N terminally truncated PilE structure, from *N. meningitidis* strain MC58, is available in the Protein Data Bank (PDB ID 4V1J). MC58 PilE is 89% identical in sequence to the SB pilin variant of strain 8013 PilE described here, differing mostly in the hypervariable region. PilE 4V1J crystallized in a different space group than ours, P6, and was solved to 1.43 Å resolution. This structure superimposes well on our strain 8013 PilE structure (root mean square deviation for main chain atoms, 1.1 Å, Supplementary Fig. 1). Small differences in the backbone

conformations occur mainly in the hypervariable region at the end of the β6 strand of the β-hairpin and in the loop following this strand. The MC58 pilin has an Ala142 in place of the very prominent Lys140 seen in our 8013 PilE structure. Electron density is absent for a single residue, Ala131, at the tip of the β-hairpin loop in 4V1J, and the β3-β4 loop conformation differs very slightly between the two pilins.

Not surprisingly, the Nm PilE structure is highly similar to Ng PilE (Fig. 2a, b), superimposing with a root mean square deviation of 3.1 Å for all main chain atoms. Since the N-terminal 54 residues are identical in both proteins the missing 29

**Table 1 | Nm ΔN-PilE data collection and refinement statistics.**

| Data collection | Nm ΔN-PilE |
|---|---|
| Beamline | SSRL 7-1 |
| Wavelength (Å) | 0.9753 |
| Space group | P2₁2₁2₁ |
| Cell $a$, $b$, $c$ (Å) | 43. 5, 46.3, 48.4 |
| Cell $α$, $β$, $γ$ (°) | 90.0, 90.0, 90.0 |
| Resolution (Å) | 1.44 |
| Completeness (%) | 98.7 (88.1)* |
| No. of observed reflections | 118934 |
| No. of unique reflections | 18088 |
| $R_{meas}$ (%) | 6 .0 (60.7) |
| $R$-factor$_{obs}$ | 5.5 (53.9) |
| $I/σ(I)$ | 21.6 (2.8) |
| Wilson $B$ value (Å²) | 18.7 |
| Mosaicity (°) | 0.5 |
| | |
| *Refinement* | |
| Resolution limits (Å) | 33.43-1.44 |
| Z | 4 |
| No. of reflections used | 18030 |
| $R_{cryst}$ (%) | 18.7 |
| $R_{free}$ (%) | 21.0 |
| | |
| *No. of non-hydrogen atoms:* | |
| Protein | 1037 |
| Water | 156 |
| | |
| *Avg B factor (Å²):* | |
| Protein | 16.5 |
| Water oxygen | 24.8 |
| RMSD bond lengths (Å) | 0.004 |
| RMSD bond angles (°) | 0.679 |
| Ramachandron plot | |
| Favoured (%) | 94.2 |
| Allowed (%) | 5.8 |
| Outlier (%) | 0 |
| PDB ID | 5JW8 |

RMSD, root mean squared deviation
*Values in parenthesis represent the highest resolution shell.

N-terminal residues in Nm ΔN-PilE monomer are also expected to be α-helical, forming a continuous α-helix with α1C, as seen in the full-length Ng PilE structure. The conserved Pro22 and Gly42 introduce kinks in Ng PilE, giving α1 a gentle S-shaped curve (Fig. 2b). Both proteins share essentially the same secondary structure elements in the globular domain (Figs 1a and 2). The only notable main chain deviations are in the position of the β3–β4 loop, which bends away from the β-strand in Nm PilE to a greater degree than it does in Ng PilE, and in the β-hairpin, which is more twisted in Nm PilE and has a two-residue insertion in its loop that allows it to contact the αβ-loop.

**CryoEM reconstruction of the *N. meningitidis* T4P.** To better understand the role of *N. meningitidis* T4P (Nm T4P) in host cell adhesion and signalling, we generated a cryoEM reconstruction. Not only are the Nm T4P filaments quite flexible, but image analysis indicated substantial variability, mainly in the twist (Methods). The Iterative Helical Real Space Reconstruction (IHRSR)[41] approach was used to obtain the reconstruction, which after sorting converged to a rise of 10.3 Å and a rotation of 100.8° for the subset of segments used. The cryoEM map, with an estimated resolution of ∼6 Å, shows well-defined density for the PilE globular domain, with a central depression and a protruding ridge on one side (Fig. 3a). Clear connectivity between the globular domains is observed between subunits in the

right-handed 4-start (+4) helix, and to a lesser extent in the left-handed 3-start (−3) and right-handed 1-start (+1) helices. Rod-like density is present in the core of the filament consistent with the protruding N-terminal α-helical segments of PilE, α1N (Fig. 3b, c). This density shows that the α1N segments are staggered in a helical array in the filament core and partially tilted relative to the filament axis, with their N termini close to the centre of the filament and their C termini lying at a larger radius. Surprisingly the α-helical density is not continuous, being considerably weaker just before it connects with the globular domains (red asterisks in Fig. 3c). This weak density indicates loss of α-helical order and increased flexibility in a central segment of α1 in the intact pilus.

**Atomic model of the *N. meningitidis* T4P.** As a first step in building an Nm T4P model, the full-length Ng PilE structure was fit as a rigid body into the Nm pilus cryoEM density map to determine a global fit for Nm PilE that included α1N. While the Ng PilE globular domain, which includes residues ∼28–53 of α1 (α1C), fits nicely into the map, the kink in α1N at Pro22 drives the N-terminal end of α1N well away from the rod-like density corresponding to this subunit (Fig. 3c). Thus, we built a model for the Nm T4P by separately fitting the Nm ΔN-PilE structure (residues 29–161) and α1N from the full-length Ng PilE structure, separated into two segments, α1:1–14 and α1:24–28, into the cryoEM map. These α-helical segments were then joined by an extended non-helical segment, α1:15–23 (Fig. 3d).

The Nm T4P model was built iteratively as described in Methods. The final model was selected based on its fit to the electron density map, minimal steric clashes between subunits, and close proximity of the Glu5 side chain with the N-terminal amine of Phe1 in adjacent subunits. Refinement statistics are shown in Table 2. The diameter of the Nm T4P reconstruction and filament model is ∼60 Å. The Nm ΔN-PilE crystal structure within the pilus model fits nicely into the cryoEM map (Fig. 4a, Supplementary Movie 1), having undergone only minor conformational adjustments during fitting and refinement including movement of the β3–β4 loop to bring it into proximity with an adjacent subunit (Fig. 4b). The ridge of density located on one side of the globular domain is occupied by the hypervariable segment of the D-region, the β-hairpin loop, β6 and the loop following β6, with Lys140 prominently displayed at its most protruding point. Lys140 is critical for pilus bundling and T4P-mediated bacterial aggregation, a phenotype associated with the high adhesive variant of *N. meningitidis* strain 8013 (ref. 42). The long axes of the globular domains run along the 4-start helix, each strand accentuated by the hypervariable ridges (Fig. 4a). The hypervariable region is implicated in binding to and remodelling of endothelial cells and was predicted based upon homology with Ng PilE to be surface-exposed[38]. Four key residues on Nm PilE mediate interactions of Nm T4P with β2AR and activation of endothelial cells: Thr130, Lys140, Asp143 and Lys144 (ref. 38). Thr130 lies on the β-hairpin β5–β6 loop near the top of the protruding hypervariable ridge whereas Lys140, Asp143 and Lys144 are located near the bottom of this ridge (Fig. 4a,b). The distance between these two binding sites on one PilE subunit (∼26 Å) is approximately equal to the distance between these sites in neighbouring subunits in the 4-start helix. Thus, β2AR may bind to individual pilin subunits but may also recognize a conformational epitope spanning two subunits.

Pilin subunits are connected in the Nm T4P by contacts between the globular domains, between the globular domains and the α1s, and between the α1s themselves. The staggered globular domains along the 1-start helix have limited contact, between the β3–β4 loop on one subunit and the αβ-loop on the next (Fig. 4c).

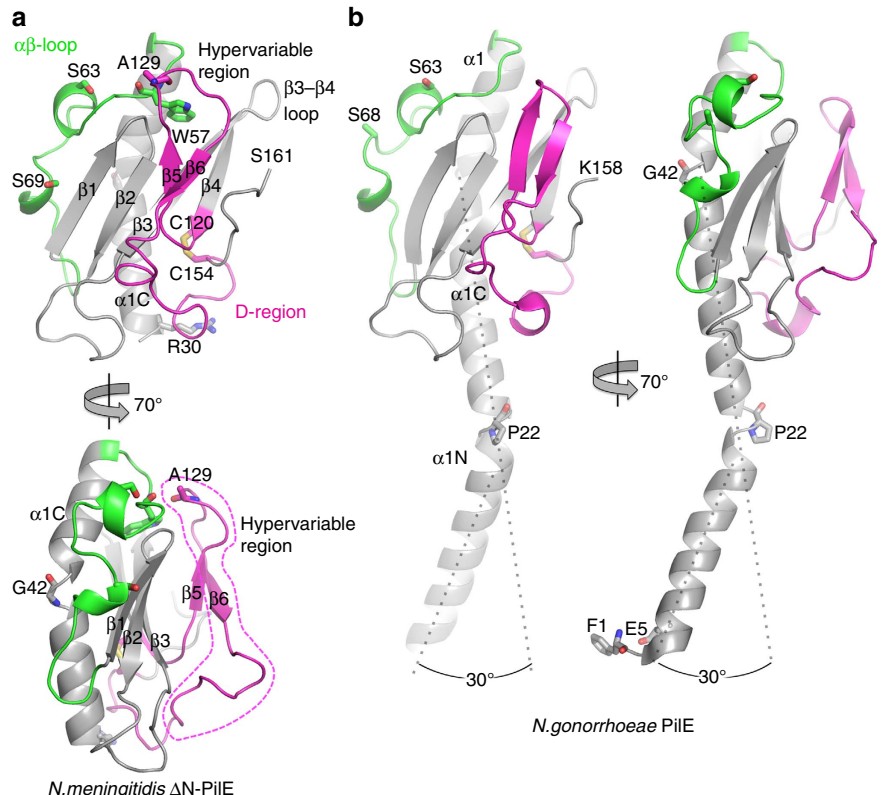

**Figure 2 | X-ray crystal structure of *N. meningitidis* ΔN-PilE and comparison with the full-length structure of Ng PilE.** Two views of (**a**) Nm ΔN-PilE and (**b**) Ng PilE pilins are shown: a 'surface view' predicted to form the outer face of the T4P filament, and a 70° rotation. The αβ loop is shown in green and the D-region is shown in magenta, with the hypervariable region within the D-region outlined in the lower panel. Secondary structures are indicated. Nm ΔN-PilE residues Ser63 and Ser69 are post-translationally modified in the native pilin, as are the corresponding Ser63 and Ser68 shown in the full-length Ng PilE structure. The Ng PilE N-terminal α-helix, α1, is kinked at Pro22 and Gly42. The N-terminal half of α1, α1N, is absent in the Nm ΔN-PilE structure, but is identical in sequence and thus expected to share the same conformation as in Ng PilE.

The bottoms of each globular domain fit into a gap between two globular domains in the next turn of the 1-start helix: the β2–β3 loop of the upper subunit contacts the tip of α1 of one of the lower subunits, and the D-region of the upper subunit contacts both the top of α1C and the αβ-loop of the other lower subunit (Fig. 4c). These interactions, mediated by mostly polar and charged residues, connect the globular domains in each consecutive turn of the 1-start helical strand, and would block antibody access the EYYLN epitope at the end of α1C. The interactions in the core of the filament are more extensive and involve mainly hydrophobic residues. The primary contact points between the α1s occur along the 1-start helical strand: the continuous α-helical segments α1:24–41, which are an integral part of their respective globular domains, alternate along the 1-start helix of the pilus filament with the ordered segments of α1N (residues 1–14) from subunits in the next turn up in the 1-start helix (Fig. 4d, e). Each α1:1–14 inserts between two α1:24–41s of the 1-start helical turn below it, contacting both α1:24–41 and the globular domain for one of the subunits and α1:24–41 alone for the adjacent subunit. The most N-terminal portion of α1:1–14 extends to the helical turn below this one to contact the C-terminal residues of α1 at the EYYLN epitope (Fig. 4e). Thus, each subunit contacts three consecutive turns of the 1-start helix—the top one via globular domain interactions, the one below that via α1:1–14 interactions with α1:24–41 and globular domains, and the one below that via α1:1–14 interactions with the EYYLN epitope of α1. The hydrophobic interactions in the α-helical core of the T4P provide the remarkable stability of the pili, which are resistant to heat and chemical denaturation and require treatment with detergent in order to dissociate them[14].

**The α-helical core of the *N. meningitidis* T4P.** In our Nm T4P model the α1 segment 24–53 is curved but runs roughly parallel to the filament axis while α1:1-14 is tilted with its N terminus oriented toward the centre of the filament (Fig. 5a–c). This orientation places the side chain of Glu5 in close proximity (<5 Å) to the N-terminal amine of Phe1 (Phe1:N) on the neighbouring α1 in the 1-start helix, consistent with a salt bridge, which matches perfectly the well-defined channel connecting the rod-like densities at their termini (Fig. 5b,c). Glu5 is invariant in Type IV pilins and Type II secretion pseudopilins and is also present in most minor (pseudo)pilins. Although the Phe1-Glu5 salt bridge has been proposed previously to neutralize these charges in the hydrophobic core of the filament[11] and to drive subunit docking into a growing pilus[9] this is the first structural evidence of such an interaction in a T4P. Neither the rod-like density for α1, nor the connection between these densities, were resolved in the 12.5 Å Ng T4P structure[9]. Thr2 is also positioned to form an intermolecular hydrogen bond with Glu5. A hydroxyl-containing threonine or serine is conserved at position 2 and is the only polar amino-acid apart from Glu5 in the first 25 residues of the Type IV pilins. Glu5 is essential for efficient T4P assembly[14,16–22] and Type II secretion pseudopilus assembly[43]. We have proposed that Glu5 facilitates docking of incoming pilin subunits into the growing pilus through electrostatic interactions with Phe1:N of the preceding subunit, neutralizing these two charges in the otherwise hydrophobic environment of the inner

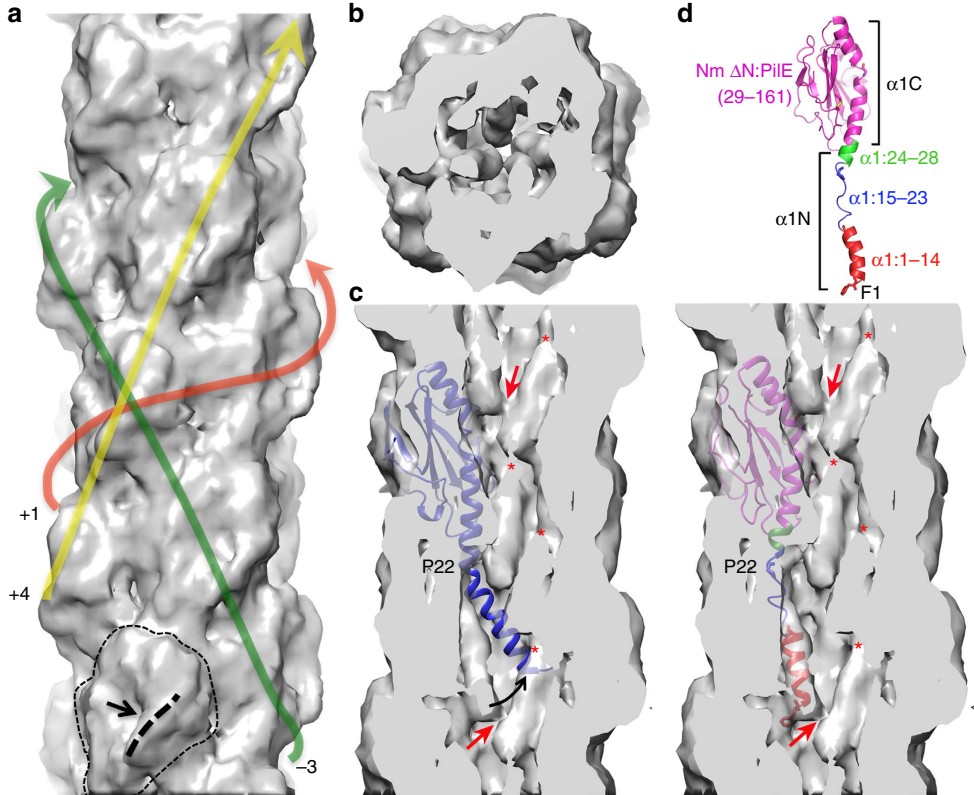

**Figure 3 | CryoEM reconstruction of the *N. meningitidis* T4P at 6 Å resolution.** (**a**) Side view of the Nm pilus cryoEM map showing the globular domain density and its connectivity in the right-handed 1-start helix (+1), the right-handed 4-start helix (+4) and the left-handed 3-start helix (−3). A single globular domain is outlined at the bottom of the map, with the ridge indicated with a thick dashed line and the central depression indicated with an arrow. (**b**) End view of a section of the cryoEM map showing the rod-like density corresponding to N-terminal α-helices. (**c**) Cross section of the Nm pilus reconstruction showing the rod-like α-helical density, the ends of which are connected near the central filament axis (red arrows). The full-length Ng PilE structure (2HI2) was fit as a rigid body into the globular domain density. The Pro22-induced kink results in the N-terminal α-helical segment completely missing the rod-like density. The rod-like density disappears just before it enters the globular domain density (red asterisks). (**d**) The Nm PilE structure was fit into the map in segments, as indicated. α1N was modelled from the corresponding region in the Ng PilE crystal structure, with the helix melted in α1:15–23, which straightens α1.

| Table 2 | Refinement statistics for the Ng T4P filament model. | |
| --- | --- |
| *RMS deviations* | |
| Bonds (Å) | 0.009 |
| Angles (°) | 1.04 |
| | |
| *Ramachandran plot* | |
| Favoured (%) | 75.5 |
| Allowed (%) | 23.9 |
| Outliers (%) | 0.6 |
| Rotamer outliers (%) | 0.0 |
| Cβ deviations | 0 |
| RMS, root mean square. | |

membrane[9] (Fig. 6a). This salt bridge appears to be maintained in the assembled pilus and may also contribute to filament stability.

The cryoEM map shows that α1 is non-helical between the helix-breaking residues Gly14 and Pro22. The loss of α-helical order at α1:15–23 compared with the all-helical conformation seen in the Ng PilE crystal structure lengthens α1 and effectively removes the Pro22-induced kink. By eliminating the α-helical structure for α1:15–23 we were able to place α1:1–14 into the rod-like EM density, at a position quite different from that

predicted for Ng T4P (Fig. 3c,d). Yet the α-helices are not particularly tightly packed, leaving gaps in the filament core and a channel ∼11 Å in diameter that winds through the filament (Figs 3b and 5a). This channel could accommodate flexibility in α1:15–23 and in the pilus filament itself.

## Discussion

Our 1.44 Å crystal structure of Nm ΔN-PilE, together with our ∼6 Å resolution cryoEM reconstruction of the Nm T4P allowed us to generate a model for this pilus that closely resembles our earlier 12.5 Å Ng T4P structure (Supplementary Fig. 2) but reveals new structural features that inform pilus biology. The non-helical conformation revealed for residues 15–23 of α1 is surprising given that this region is α-helical not only in the Ng PilE monomer structure but also in the other full-length Type IV pilin structures from *Pseudomonas aeruginosa*[8], *Dichelobacter nodosus*[10] and *G. sulfurreducens*[12] all of which possess S-shaped α-helices with kinks at Pro22 and Gly/Pro42. These full-length Type IV pilin structures were obtained by dissociating purified pili into pilin subunits with the detergent octyl β-D-glucopyranoside (βOG), which disrupts hydrophobic interactions and solubilizes α1N. In bacteria α1N is anchored in the inner membrane prior to pilus assembly[13]. The α-helical conformation would shield the polar nitrogens and oxygens in the α1N backbone from the hydrophobic phase of the lipid bilayer

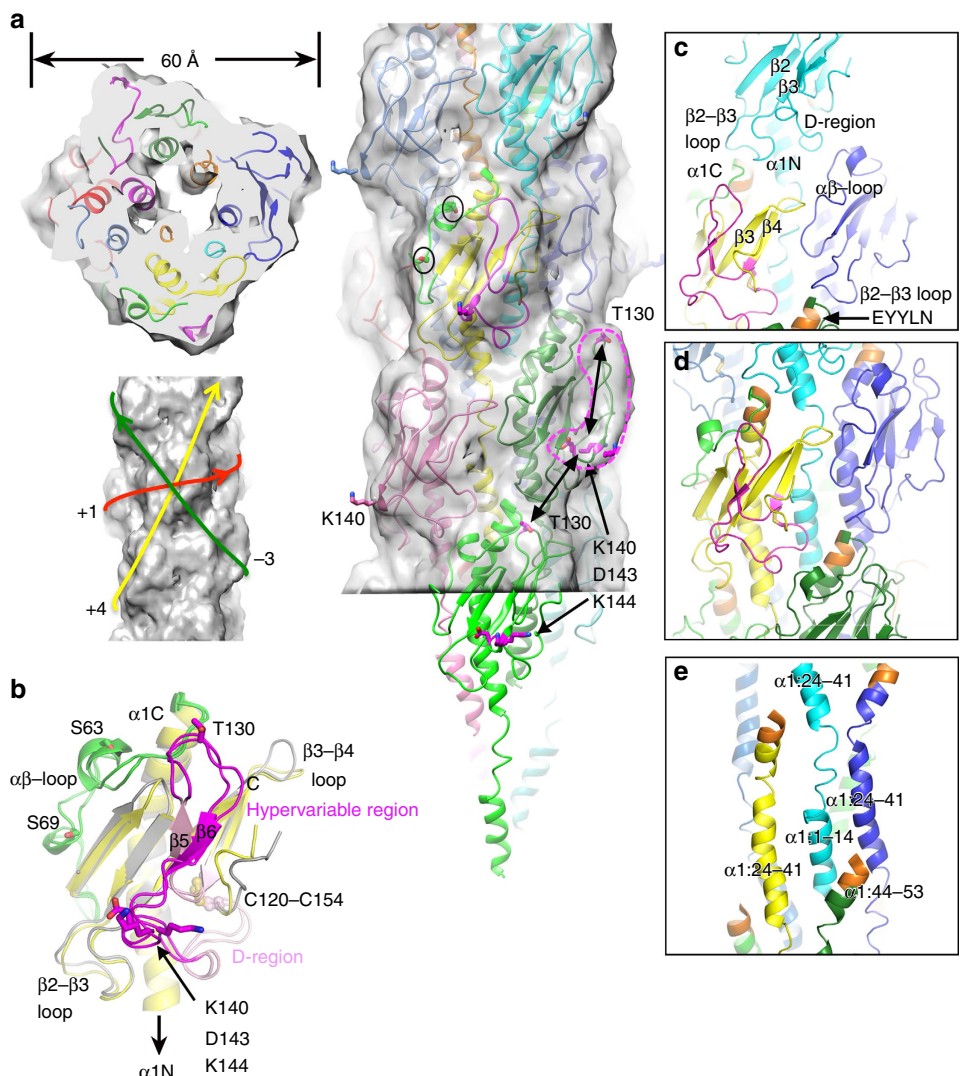

**Figure 4 | Structure of the *N. meningitidis* T4P.** (**a**) End and side view of the Nm T4P model fit into the cryoEM density, with each subunit shown in a different colour. In the side view the central subunit is coloured yellow with the αβ-loop (residues 54–79) in green and the D-region (120–154) in magenta. Circled residues Ser63 and Ser69 within the αβ-loop are post-translationally modified in the native T4P but these modifications are not resolved in the cryoEM reconstruction. The hypervariable region (127–146) within the D-region forms a ridge on the filament surface, indicated by the broken magenta line for one subunit. Residues Thr130, Lys140, Asp143 and Lys144 within the hypervariable ridge in this subunit and the adjacent green subunit are shown in stick representation with carbon atoms coloured magenta. These two β2AR interaction sites are spaced equidistant apart (∼26 Å) within a subunit and between subunits, as indicated by the double-headed arrows. The connectivity of the subunits is shown on the left below the end view. (**b**) Superposition of Nm ΔN-PilE before (grey, green and magenta) and after refinement in the pilus model (yellow, green and magenta). The D-region is coloured light pink and the hypervariable ridge within the D-region is coloured magenta. (**c**) Globular domain interactions shown from the perspective of the central subunit coloured as in (**a**), with the EYYLN epitope coloured orange. (**d**) The globular domains and their corresponding α1:24–41 segments follow the +1-start helix and sandwich between them α1:1–14 from subunits in the preceding turn of the +1-start helix. (**e**) Interactions among α1s as shown in **d** but with residues 54–161 of the globular domains removed for clarity. α1 segments are labelled.

(and also from the acyl chains of βOG during crystal growth). Thus, the melting of the helix at α1:15–23 appears to be induced on integration of the pilin subunits into the pilus filament, perhaps to facilitate packing of α1N (Fig. 6a). The conserved helix-breaking residues Gly14 and Pro22 may destabilize the α-helical secondary structure in this region, allowing it to unfold during pilus assembly. Interestingly the Type II secretion pseudopilins also have Gly14 and Pro22 (Fig. 1b) and thus may also adopt a non-helical conformation in this region within the pseudopilus. Similarly, although the Type IVb pilins lack the proline at position 22 they have glycines at positions 11, 14 and 19, which may destabilize this region in a manner similar to that of Gly14/Pro22. In contrast, archaeal flagellins contain an N

terminus that is homologous to α1N in the Type IV pilins[44] but possess no proline in this segment, and glycines are not in conserved positions (Fig. 1b). A ∼ 4 Å resolution cryoEM reconstruction of an archaeal flagellar-like filament from *Igniococcus hospitalis* was recently determined, which shows well-defined density corresponding to a straight, fully α-helical α1N (ref. 45). Thus, the loss of helical order seen for Nm T4P may be a shared feature of the T4P and Type II secretion pseudopili but is not necessarily a universal feature of filament-forming proteins containing a Type IV pilin N-terminal domain.

The new Nm T4P model with its non-helical α1 segment provides a molecular basis with which to understand the reversible force-dependent polymorphism observed for both

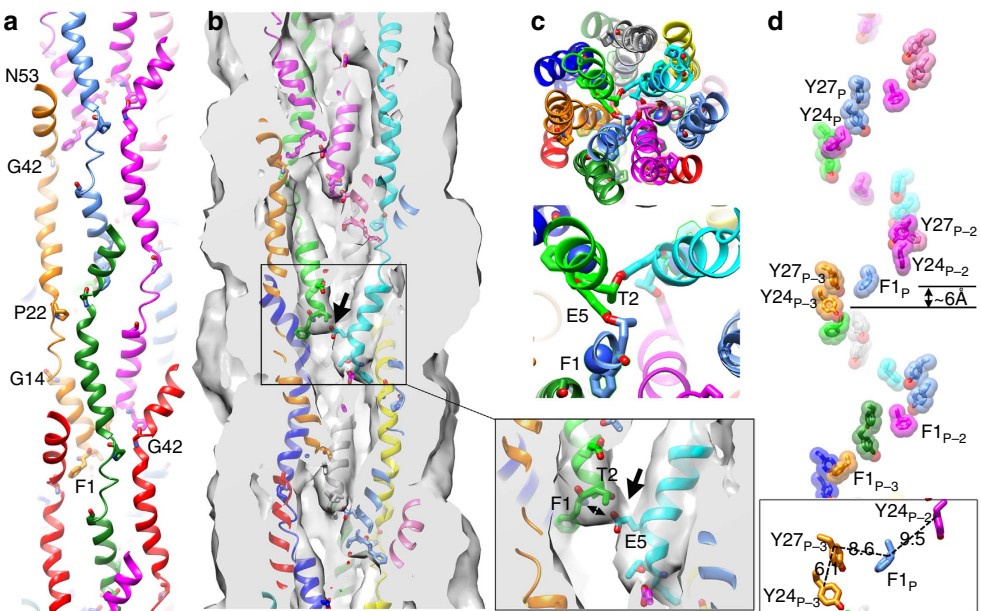

**Figure 5 | Conformation and interactions among the N-terminal α-helices of the Nm T4P.** (**a**) Side view of the α1 segments only (residues 1–53) from the outside of the pilus, with Phe1, Glu5, Gly14, Pro22 and Gly42 shown in stick representation. (**b**) Cross-section of the α1s in the cryoEM map. The interaction between the Glu5 side chain on one subunit and Phe1:N and the Thr2 side chain in the adjacent subunit in the 1-start helix is shown enlarged. The arrow points to the electron density corresponding to these contacts. The Phe1 side chain interacts with α1C in the globular domain of an adjacent subunit. (**c**) End view of the α1 core with a close-up of the interactions between Glu5 and Thr2 and Phe1:N (shown as a blue sphere). (**d**) Positions of the aromatic side chains of conserved residues Phe1, Tyr24 and Tyr27, all on α1. Residues are numbered according to the position of their corresponding subunit or protomer, Pn, relative to the central subunit P (light blue). The Phe1 side chain is located between Tyr24/27 pairs on adjacent α1s in the +1 helix, resulting in a continuous band of aromatic side chains that winds up the filament axis. These residues are implicated in electron transfer for the conductive *G. sulfurreducens* T4P[48,49]. Cγ–Cγ distances for the final Nm T4P model are shown in the inset, although their precise positions are not known with certainty due to the limited resolution of the reconstruction.

Nm and Ng T4P[29,32]. Ng T4P can be stretched to approximately three times their length when subjected to forces of ~ 100 pN, with a corresponding narrowing of the filament diameter[29]. This force-induced conformational change exposes the EYYLN epitope at the end of α1, which is normally buried in the pilus and only accessible at the tip (Figs 4c–e and 6). Similar exposure of this epitope is induced for Nm T4P upon binding to endothelial cells[32]. The mechanism by which Nm and Ng T4P can extend in a reversible manner was not apparent from the lower resolution Ng T4P model, in which the α1s are all-helical and twisted around each other in the filament core[9]. The new Nm T4P model can be envisioned as a coil-like structure: each turn of the coil, which follows the 1-start helical path, is comprised of the globular domains connected by a small number of electrostatic/polar interactions and by extensive hydrophobic interactions with intervening α1Ns (α1:1–14) from the subunits in the next turn of the coil; each consecutive turn of the coil is held together axially by electrostatic and polar interactions between the globular domains and by α1, which acts as a tether between each turn (Figs 4d,e and 6b, left panel). Shear forces on the pili may disrupt the weaker axial interactions between globular domains in consecutive turns of the coil, and extend residues α1:15–23, stretching the pilus while maintaining the lateral interactions between globular domains, α1:24–41 and α1:1–14 along the coil (Fig. 6b, right panel). Thus, force would produce a spring-like motion separating the coils to elongate the pilus. Such a motion was described recently for the P pilus of the donor chaperone pilus family based on the 3.8 Å resolution cryoEM reconstruction[46]. The architecture of the P pilus is markedly different from that of the T4P: elongated P pilin subunits are connected along the coils by donor-strand exchange of their N-terminal extensions, which insert into a groove in the next

subunit to complete an Ig fold[47]; polar interactions hold the coils of the P pili together in the axial direction. Yet P pili, like the Nm and Ng T4P, can extend under force in a reversible spring-like manner, providing an interesting example of convergent evolution.

In our Nm T4P model α1:15–23 spans ~ 18 Å. In a fully extended conformation these nine residues could potentially span ~ 32 Å, which would take the axial rise per subunit from 10.3 Å to ~ 24 Å, an increase of ~ 230%. The melting of a few residues flanking 15–23 could further increase the filament length. Such an extension might separate the globular domains between consecutive turns of the coil enough for antibody to access to the EYYLN epitope, while maintaining the lateral hydrophobic interactions along the coil, between the globular domains, α1:24–41 and α1:1–14. Whereas the accompanying increase in rise per subunit can occur in the P pili by simply tilting the subunits, the more complex interactions holding the Type IV pilins may also shift relative to each other, changing the register of the α1 interactions to increase the rise per subunit, transiently resulting in thinner longer pili. Upon release of force, α1N would collapse to the more compact and stable conformation seen in the Nm T4P model and interactions between the coils would reform. This conformational elasticity would allow *N. meningitidis* to remain attached via T4P to endothelial cells in the brain vasculature, and *N. gonorrhoeae* cells to remain attached to the lining of the urethra under shear forces from blood and urine flow, respectively. Of note, the force-induced stretching of Nm T4P would separate the β2AR-binding residue Thr130 from the Lys140/Asp143/Lys144 patch in adjacent subunits (Fig. 4b). Such a change may provide a signal to activate β2AR and cortical plaque formation, leading to loosening of the endothelial cell junctions and opening of the blood brain barrier. The conformational flexibility of the Nm T4P is likely

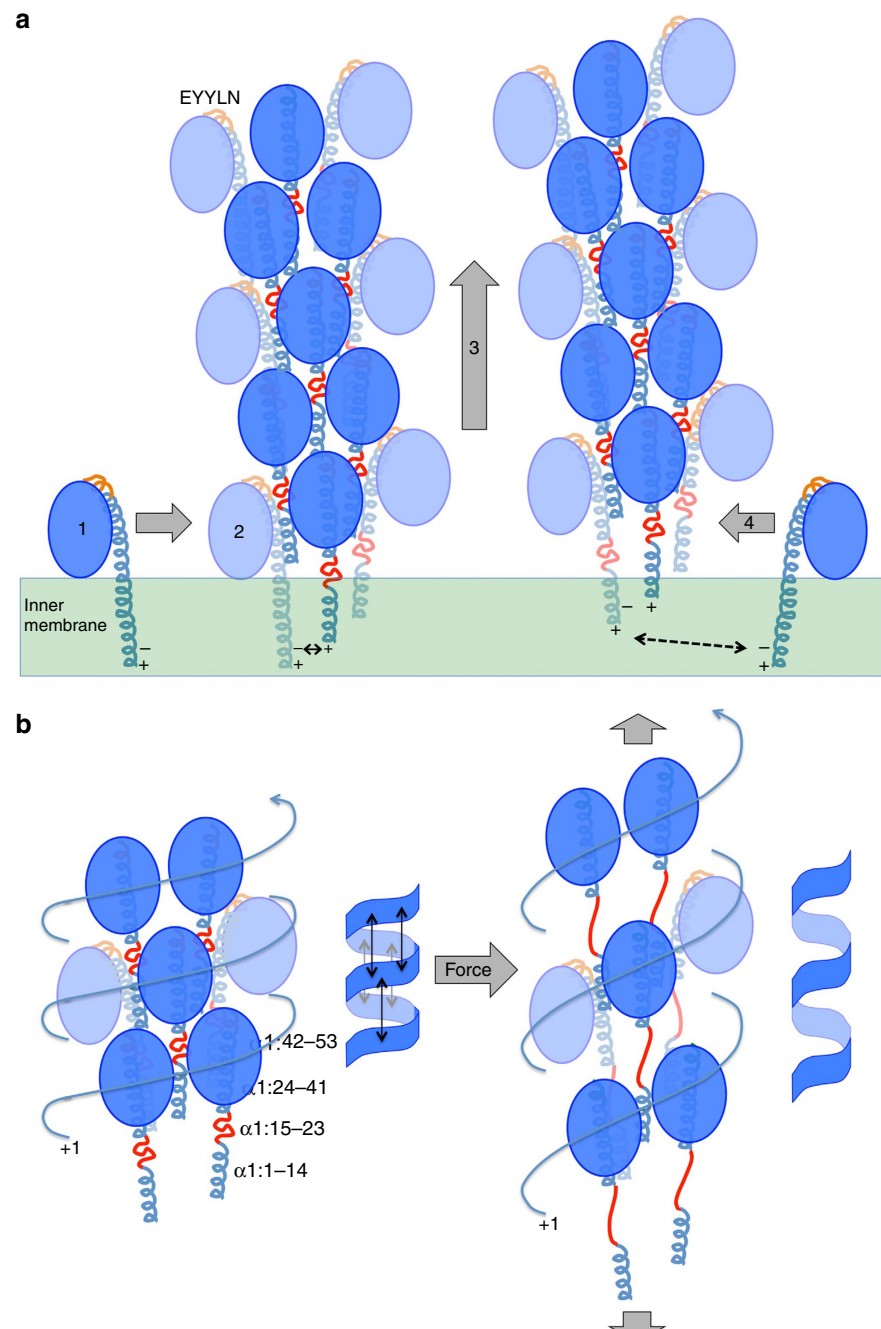

**Figure 6 | Models for Type IV pilus assembly and force-induced conformational change. (a**) T4P assembly model. (1) Before their incorporation into pilus filaments pilin subunits are anchored in the inner membrane with α1 in an all-helical conformation to shield backbone oxygens and nitrogens from the acyl phase of the lipid bilayer. (2) Pilin subunits dock into a growing pilus filament, attracted in part by charge complementarity between Glu5, represented by the ( − ) charge, and the positively charged N-terminal amine ( + ) on the terminal subunit in the growing pilus. (3) The pilus filament is extruded out of the membrane a short distance by the pilus assembly machinery, opening up a gap at the base of the pilus for a new subunit to dock. As the newly added pilin is extruded out of the inner membrane α1:15–23 (red) melts in order for α1:1–14 to pack into the filament core in an α-helical conformation. (4) Another subunit docks into the gap at the base of the pilus. The EYYLN epitope at the top of α1 is shown in orange. This epitope is exposed at the tip of the pilus and buried along its length. (**b**) Model for the force-induced conformational change of Nm and Ng T4P. Left panel: in its relaxed T4P conformation subunits are held together by lateral interactions between the globular domains along the +1-start helix and alternating α1Ns (α1:1–14) from the next turn up, as well as by axial interactions among the globular domains in consecutive turns of the +1-start helix (represented by the double-headed arrows in the schematic of the coil). The central portion of α1, α1:15–23 (red), is non-helical but compact. The EYYLN epitope (orange) is only exposed at the tip of the pilus. Right panel: under stress, the axial interactions are disrupted and α1:15–23 becomes fully extended but the lateral interactions between the globular domains and α1:1–14 are maintained for the most part, allowing a spring-like extension of the pilus to expose the EYYLN epitope all along its length.

responsible for the structural heterogeneity observed during processing of the cryoEM images, which precluded a higher resolution reconstruction.

The Nm T4P structure may also be relevant in considering the mechanism for charge transfer in the related T4P from *G. sulfurreducens*. The *G. sulfurreducens* Type IV pilin is unique

in its absence of a globular domain: this 61 amino-acid pilin has an extended curved α1, as seen for Ng T4P, followed by an unstructured 8 amino-acid C terminus[12]. This pilin shares 50% identity with Nm PilE in its N-terminal 30 amino acids, including Gly14 and Pro22 (Fig. 1b), thus it may also have a non-helical α1 segment when incorporated into a pilus. The *G. sulfurreducens* pilin contains 6 aromatic residues, Phe1, Tyr24, Tyr27, Tyr32, Phe51 and Tyr57, five of which are also present in Nm PilE. These residues are implicated in charge conduction in *G. sulfurreducens* T4P, as a mutant, Aro-5, with alanine substitutions for all aromatics except Phe1, makes pili but shows a 10-fold reduction in conductivity[36,48,49]. The authors proposed metallic-like charge conductivity through closely spaced π–π stacked aromatic side chains, similar to electron transport through polyaniline or polyacetylene nanostructures[48]. This model is supported by X-ray microdiffraction data for wild type conductive *G. sulfurreducens* T4P, which show a 3.2 Å peak representative of a repeating feature within the pilus; this peak is absent in the Aro-5 mutant[35]. Since the spacing of the aromatic residues in existing T4P models[8,9,14] is considerably larger than 3.2 Å, new computational models for the *G. sulfurreducens* T4P have been proposed that bring the core aromatic residues Phe1, Tyr24 and Tyr27 within π–π stacking distance (3–4 Å) to facilitate long-range electron transport through the pilus[35,50]. Such energy-minimized models have a comparable rise to Ng T4P (∼ 10 Å) but a much smaller twist (55–60°) and are more compact[50]. While feasible, these models assume an all-helical α1. In our Nm T4P model with its non-helical α1 segment, the aromatic residues are not within π–π stacking distance, with most being > 8 Å away (Cγ–Cγ) from their closest neighbour. Phe1, Tyr24 and Tyr27 form an aromatic band that spirals up the filament axis (Fig. 5d), whereas residues 32, 51 and 57, corresponding to the remaining aromatic amino acids in the *G. sulfurreducens* pilin, occupy more peripheral positions. Phe1 lies at approximately the same level along the Nm T4P filament axis as the Tyr27 of three subunits preceding (protomers P and P-3, respectively) and ∼ 6 Å from Tyr24 (Fig. 5d), distances that are incompatible with π orbital overlap and the 3.2 Å diffraction peak seen for *G. sulfurreducens* T4P. The spacing of the aromatic side chains is constrained in Nm and Ng T4P by the presence of the globular domain, which places a lower limit on the subunit rise and rotation. The absence of a globular domain for the *G. sulfurreducens* Type IV pilins means that they can pack more closely in the pilus, with smaller rise and rotation values. These parameters, together with a non-helical, partially extended portion of α1, may provide more realistic *G. sulfurreducens* T4P computational models that aromatics except Phe1, Tyr24 and Tyr27, and perhaps other aromatic side chains, within 3-4 Å of each other to form a continuous chain of π–π stacked aromatic side chains for charge transfer. Interestingly, reduction of the solution pH from 10.5 to 2 results in a 100-fold increase in electrical conductivity for *G. sulfurreducens* T4P and a corresponding increase in the intensity of the 3.2 Å diffraction peak[35]. A possible explanation for this apparent increase in order is that the low pH disrupts the Phe1-Glu5 salt bridge, freeing the Phe1 side chain to optimize its orientation for charge transfer.

In conclusion, the 6 Å structure of the Nm T4P provides a framework for understanding Nm pilus interactions with host cell receptors as well as more general aspects of T4P biology including assembly, function and biophysical properties. The highly conserved T4P N-terminal domain is all-helical in X-ray crystal structures of full-length Type IV pilins, which most likely represents its state in the inner membrane. On the basis of these crystal structures and low resolution EM reconstructions it has been assumed until now that α1 is all α-helical in the intact pilus as well. We have shown that the N-terminal segment is actually partially melted in the assembled pilus. The melting of this region helps to explain pilus heterogeneity, flexibility and elasticity, including T4P's ability to stretch under force, and in the case of *G. sulfurreducens* T4P, to conduct charge. N-terminal sequence similarity between Nm PilE and other Type IV pilins and Type II secretion pseudopilins suggest that this α1N melting may be a conserved feature, yet one that does not extend to the archaeal flagellins. The Nm T4P cryoEM reconstruction shows the limitations of secondary structure predictions, which can never be more than 80–90% accurate as they fail to account for contextual influences on protein structure. It also illustrates the value of obtaining the highest resolution possible for macro-molecular assemblies, as key structural features can be missed with rigid body fitting of atomic structures of isolated subunits. Such features can have important biological implications for the protein complexes.

## Methods

**N. meningitidis PilE expression and purification.** The gene fragment encoding PilE residues 29–161 (ΔN-PilE) was cloned from genomic DNA of *N. meningitidis* 8013 (high adhesive strain, SB)[37] using the primers Nm-pilE-EcoRI-fpcr (5′-CGG AATTCGCCCGCGCACAAGTTTCCGAAGCCATCCT-3′) and Nm-pilE-PstI-rpcr (5′AACTGCAGTTAGCTGGCAGATGAATCATCGCGGCAGGT-3′) and inserted into the pMAL-p2x plasmid (NEB) at the EcoR1 and PstI cloning sites. The *pilE* gene was transferred into the pET15b vector (Novagen) restriction sites Nde1 and BamH1 using primers PilE-FP (5′-GCAATTCCATATGGCCCGCGCA CAAGTTTCC-3′) and PilE-RP (5′-CCGCGGATCCTTAGCTGGCAGATGAATC ATCGC-3′). pET15b-*pilE*, which encodes an N-terminal His-tag, was transformed into *E. coli* BL21(DE3) SHuffle strain (NEB) and grown at 16 °C for 22 hrs in LB, 0.4 mM IPTG. Cells were lysed by sonication in lysis buffer (20 mM Tris-HCl, pH 8, lysozyme, 1 mg ml$^{-1}$, EDTA-free protease inhibitor (Roche)) and cell debris was removed by centrifugation. The lysate was loaded onto a Ni-NTA column and bound His-ΔN-PilE was eluted using elution buffer (50 mM Tris-HCl, pH 8, 100 mM NaCl, 250 mM imidazole) then dialysed in the presence of thrombin to simultaneously remove the imidazole and the His-tag. ΔN-PilE was further purified by size exclusion chromatography using a HiPrep 26/60 Sephacryl S-100 HR column (GE Healthcare) in buffer containing 20 mM Tris-HCl, pH 7.4, 100 mM NaCl. Fractions containing ΔN-PilE were pooled, concentrated to 15 mg ml$^{-1}$ using a stirred-cell concentrator (Millipore) and flash-frozen.

**ΔN-PilE crystallization, data collection and refinement.** ΔN-PilE crystals were grown by the hanging drop vapour diffusion method at 293 K. Initial crystallization conditions were obtained from the high throughput screening laboratory at the Hauptman-Woodward Medical Research Institute[51] then optimized in-house. Diffraction-quality crystals grew in 4 μl drops containing a 1:1 ratio of protein (15 mg ml$^{-1}$) and reservoir solution (0.1 M ammonium sulfate, 0.1 M MES pH 6, 34% (W/V) PEG 4,000). The needle shaped crystals were cryocooled in mother liquor with 30% (v/v) glycerol for storage and transport to the Stanford Synchrotron Radiation Lightsource (SSRL) for X-ray diffraction data collection.

Diffraction data for ΔN-PilE were collected at the SSRL Beamline 7.1 by remote access using Blu-Ice software[52]. Raw data were processed and scaled and structure factor amplitudes were determined by data processing suite XDS[53]. Matthews coefficient calculations using the CCP4 program MATTHEWS_COEF[54,55] indicated that the asymmetric unit contains one ΔN-PilE molecule with 39.3% solvent content. The crystal structure was solved by the molecular replacement method using the program PHASER[56] in CCP4i[54,57]. The Ng PilE crystal structure (PDB ID 2HI2[9]), with the N-terminal 29 residues removed, was used as the model. Molecular replacement yielded an unambiguous solution with z-scores RFZ = 9.8, and TFZ = 14.5. The model residues were changed to those of Nm PilE in COOT[58] where necessary and rigid body refinement was carried out followed by 20 cycles of restrained refinement using REFMAC[59], which brought $R_{work}$ and $R_{free}$ to 0.271 and 0.317, respectively. The ΔN-PilE model was examined and adjusted in COOT using difference maps. The modified model was further refined using the 'Improvement of maps by atoms update and refinement' tool in ARP/wARP[60], bringing the $R_{work}$ to 0.197 and the $R_{free}$ to 0.266. A composite annealed omit map was calculated using CNS[61] and all the residues and water oxygens were validated with the omit map, to give $R_{work}$ of 0.211 and $R_{free}$ of 0.238. At this point the coordinates were validated and deposited in the protein data bank (PDB ID 4XNP). However, after employing the PHENIX suite[62] to refine the Nm T4P model we used this program to re-refined the ΔN-PilE structure with options 'Automatically add hydrogens to model' and 'Update waters', reducing the R-factors to 0.187 ($R_{free}$) and 0.210 ($R_{free}$). These ΔN-PilE coordinates were deposited to the PDB under the accession number 5JW8. Data collection and refinement statistics are reported in Table 1.

**Preparation of N. meningitidis T4P for cryoEM.** *N. meningitidis* 8013 cells were grown on GCB plates[63], resuspended in solubilization buffer (20 mM

ethanolamine, pH 10.5) and vortexed to shear the pili from the cells. Cells were removed from the sheared sample by two rounds of centrifugation at 4,000g and pili were precipitated by the addition of ammonium sulfate to 10% saturation and 16 h incubation at 16 °C. Pili were collected by centrifugation (13,000g, 60 min, 4 °C) and resuspended in solubilization buffer. A second round of purification was performed by addition of ammonium sulfate to 10% saturation and precipitation. The final protein concentration was 0.4 µg µl in 20 mM ethanolamine, pH 10.5.

**CryoEM and image processing.** Samples (2 µl) were applied to plasma-cleaned lacey carbon grids and vitrified using a Vitrobot Mark IV (FEI, Inc.). Images were collected at an operating voltage of 300 keV on an FEI Titan Krios equipped with a Falcon 2 direct electron detector. The magnification used provided a sampling of 1.05 Å per pixel. A total of 2,295 images were collected from a single grid using the EPU automated system on the Krios, and these were reduced to 541 images after discarding those with no filaments or extensively aggregated filaments, poor contrast transfer function (CTF) and so on. Images were stored containing seven 'chunks', where each chunk represents a set of frames corresponding to a dose of $\sim 20$ electrons per Å$^2$. The full dose image stack was used for the estimation of the CTF, using CTFFIND3 (ref. 64), as well as for boxing filaments using the e2helixboxer routine within EMAN2 (ref. 65). The SPIDER software package[66] was used for almost all other operations. Phase reversals in the images were corrected by multiplying each image by the calculated CTF, which is a Wiener filter in the limit of a very low SNR.

Approximately 100,000 overlapping pilus segments, each 384 px long and shifted by 15 px ($\sim 1.5$ times the axial rise per subunit) were cut from the long boxes, and these were reduced to 69,621 after discarding those with a large out-of-plane tilt. The IHRSR method[41] was used to generate a global reconstruction, and this was then deformed corresponding to different values of the axial rise and rotation (twist) of the subunits to generate an ensemble of structures used for multi-reference sorting. This sorting suggested that most structural variability existed within the twist, and a new sorting was done keeping the axial rise fixed at 10.3 Å with a twist ranging from 98.8 to 102.8°. The central bin from this sorting, corresponding to a rotation of 100.8°, contained 15,586 segments which were used for the final reconstruction. The amplitudes of the reconstruction were corrected for the CTF by dividing them by the sum of the squared CTFs, since they had been multiplied by the CTF twice: once by the microscope, and once computationally in the phase corrections. The reconstruction was filtered to 5.0 Å and sharpened with a negative B-factor of 150 for preparation of figures.

**Atomic model of the Nm T4P.** The Nm ΔN-PilE structure (residues 29–161) was docked into the pilus reconstruction using the 'Fit in Map' tool in Chimera[67]. Residue positions in the tight loop at the C-terminal end of the αβ-loop (73–78) and the β3–β4 loop (112–115) were adjusted to optimize their fit in the map. Next, the full-length Ng PilE crystal structure was superimposed on Nm ΔN-PilE to obtain coordinates for part of the protruding segment of α1, α1N. This superposition provided coordinates for α1N residues 24–28 but residues N-terminal to these fit poorly into the cryoEM map due to the kink induced by Pro22, which directs most of α1N away from the rod-like density in the filament core. COOT[58] was used to fit varying lengths of the most N-terminal portion of the Ng PilE structure into the rod-like density (for example, residues 1–12, 1–13, 2–13, 2–14 and so on). The angle of rotation along the helical axis of this N-terminal α-helical segment was varied to minimize the distance between the Glu5 side chain and the Phe1:N in adjacent subunits in the $+1$ helix. Residues linking this α-helical segment to α1:24–28 were built in an extended conformation due to the weak density in this region. In models where residues 1 and 2 were included in the rod-like density, ideal α-helical phi/psi angles were imposed using COOT as these residues are extended in the Ng PilE structure. Bond lengths and angles of this full-length Nm PilE subunit starting model were idealized using REFMAC[59]. This PilE model, docked into the EM density, was designated Subunit A. A second PilE model was fit into the adjacent subunit density and designated Subunit B. The symmetry operation (rise and rotation) relating these two subunits was determined by LSQKAB in CCP4, and used to generate Nm T4P models of 10 or 21 subunits with CCP4 PDBSET. The cryoEM map in BRIX format was converted into CCP4 format using MAPMAN, and the Nm T4P models were refined against this map using the PHENIX suite[62]. B-factors were fixed during refinement due to the limited resolution of the reconstruction. Refined models were examined within the cryoEM map in Chimera to identify steric clashes between subunits, which were removed by altering side chain conformations and then additional refinement cycles were run. The modelling proceeded iteratively, with each new model examined in CHIMERA, modified to improve its fit in the map, minimize steric clashes and the Phe1:N-Glu5 distance followed by refinement cycles. Refinement statistics are shown in Table 2.

The resolution of the final reconstruction was estimated at $\sim 6$ Å by Fourier Shell Correlation (FSC) between the map and the atomic model (Supplementary Fig. 3). This avoids several problems with the traditional map:map FSC, such as the fact that it can be completely artifactual for helical structures[68].

**Data availability.** The Nm ΔN-PilE was deposited in the PDB under accession number 5JW8. The Nm T4P reconstruction was deposited in the Electron Microscopy Data Bank under accession number EMD-8287, and the corresponding atomic filament model was deposited in the PDB under accession number 5KUA. The data that support the findings of this study are available from the corresponding authors on request.

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

## Acknowledgements

We thank Grace (Guixiang) Yang for expression and purification of ΔN-PilE and Pavel Afonine (Lawrence Berkeley National Laboratory) for assistance with Phenix during the construction of the Nm T4P model. This research was funded by CIHR grant MOP125959 and NSERC grant RGPIN312152 to L.C. and NIH EB001567 to E.H.E. The laboratory of XN is supported by INSERM, CNRS, Université Paris Descartes, the Agence Nationale de la Recherche and the Fondation pour la Recherche Médicale.

## Author contributions

S.K. solved the ΔN-PilE crystal structure and generated and refined the Nm T4P model. M.C. and X.N. initiated the collaboration, cloned the *pilE* gene into pMal and purified Nm T4P for cryoEM. X.Y. performed the cryoEM and filament selection. E.H.E. performed the image analysis and T4P reconstruction and contributed to T4P model building. L.C. coordinated the project and guided the model building. L.C., S.K. and E.H.E. analysed the structures. E.H.E. and L.C. co-wrote the manuscript.

## Additional information

**Competing financial interests:** The authors declare no competing financial interests.

