## [Peer Review File · Nature Communications]

Reviewers' comments:

Reviewer #1 (Remarks to the Author):

Neisseria meningitidis is a bacterial pathogen responsible for life-threatening diseases such as meningitis. The organism can cross the blood brain barrier (BBB) via the interaction of its Type IV pili (T4P) with the host endothelial cell receptor CD147 and the $\beta 2$ adrenergic receptor. PilE is the major component of T4P. The PilE protein is a ladle shaped molecule with a globular domain and a long N-terminal helix $\alpha 1$. The mechanism of assembly of PilE and other pilin proteins to form the T4P, and the mode of interaction of T4P with host receptors is currently unknown. In the paper by Kolapan et al, the authors report the X-ray crystal structure of an N-terminally truncated form of the major pilin PilE protein from *Neisseria meningitidis* strain 8013 (ΔN -Nm PilE) at 1.44 Å resolution as well as a cryo-EM reconstruction of the intact T4P estimated at ~ 6.0 Å resolution. A model is derived from the cryo-EM map using the crystal structure of full length Ng PilE and their ΔN -Nm PilE structure. The crystal structure and cryo-EM model is compared to the previously published crystal structure of full length PilE from *Neisseria gonorrhoeae* (Ng PilE), and the previously published ~ 12.5 Å cryo-EM model of Ng T4P. Nm PilE has 78% overall sequence identity to Ng PilE and is 100% identical in the N-terminal helix $\alpha 1$.

Key findings from the structural studies as presented by the authors are:

1. The crystal structure of the ΔN -Nm PilE is similar to the previously reported structure of Ng PilE. A 3.1 Å r.m.s.d is reported between the two structures for main chain atoms. The structural differences are limited to the $\beta 3$ - $\beta 4$ loop that bends further away from a β -strand (which β -strand?) in ΔN -Nm PilE as compared to Ng PilE. and there are differences in a β -hairpin loop (which loop? - it was not specified). The value of the determined similarities and differences to understanding the biology should be clearly defined.
2. Their ΔN -Nm PilE structure is also similar (1.1 Å r.m.s.d for main-chain atoms) to the 1.43 Å Nm PilE structure from strain MC58. The two Nm PilE protein variants share 89% sequence identity. Minor conformational differences are noted in the text, but not described in any detail. Is there significance to the similarities or the minor differences?
3. A cryo-EM map was generated for the Nm T4P with an estimated resolution of ~ 6 Å. Rigid body fitting of the Ng PilE structure, followed by fitting in their ΔN -Nm PilE structure results in a model that shows the rod-like assembly of the T4P filament, as previously reported for Ng T4P. Well-defined density is observed for the PilE globular domain with ridge-like features, similar to those observed previously for Ng T4P. What is the new impact for the biology from this?

4. A notable difference between the Ng and Nm T4P is that the α -helical density appears to be weaker at the junction close to the globular domain and this is interpreted as loss of helical order and increased flexibility of $\alpha 1$ in T4P.
5. A solvent channel that is approximately 11 Å in diameter is observed through the T4P filament.
6. The side chain of Glu5 in $\alpha 1$ is in close proximity to the Phe1 NH₂ and side-chain of Thr2 of $\alpha 1$ of a neighboring molecule, suggesting that a salt-bridge interaction may help stabilize the T4P assembly. This interaction was previously proposed to occur in the Ng T4P cryoEM model.
7. A stacking interaction is also observed between Phe1 of one subunit and Tyr24 and Tyr27 of the second subunit, suggesting that the π - π stacking also stabilizes the overall fold.
8. Residues 15-23 are reported to be flexible in the cryo-EM model of Nm T4P, a notable difference in the Nm PilE as compared to other PilE structures and models.

Despite the crystal structures available for PilE proteins and the cryo-EM model of the Ng T4P (at lower resolution), the paper has sufficient novelty for a high impact publication, but the biological significance needs to be clarified. The *Neisseria meningitidis* Type IV pilin and pilus structures together are of considerable biological importance. The Authors should consider the concerns/comments are listed below.

Major comments:

1. There is little biological insight to be gained from the structures as currently presented in the paper. For example, on page 9, the authors state that "This structure provides a molecular framework for understanding the coordinated action of PilE, PilV, and PilC..."; yet, it is not clear what new insight has been gained about the mechanism of assembly of T4P, that was not known from other studies. Is there relevance to interpreting the movements defined from the related structures (Hexameric structures of the archaeal secretion ATPase GspE and implications for a universal secretion mechanism, EMBO J. 2007 Feb 7;26(3):878-90; Structure of an essential type IV pilus biogenesis protein provides insights into pilus and type II secretion systems. J Mol Biol. 2012 May 25;419(1-2):110-24). Structure of an essential type IV pilus biogenesis protein provides insights into pilus and type II secretion systems.). A model for assembly and disassembly of the pili should be presented based on the new and existing data, taking into account the observed flexibility of the helix $\alpha 1$. How does this model relate to the assembly proposed from antibodies? This would seem relevant for a human virulence factor, i.e. Assembly and antigenicity of the *Neisseria gonorrhoeae* pilus mapped with antibodies. Infect Immun. 1996 Feb;64(2):644-52.
2. The authors should comment on whether they tried to crystallize full-length Nm PilE. What was the rationale for crystallization of the N-terminally truncated PilE protein? Is there

structural information available from the related Nm PilE strain MC58 for the N-terminal portion of $\alpha 1$ that would corroborate the flexibility observed in T4P for this region?

3. The authors discuss several specific side chain-side chain interactions such as salt-bridges and aromatic stacking interactions in the paper. At 6 Å resolution, such interactions may be postulated or inferred, but can they be stated to exist with certainty? For example, the words "direct evidence of a salt bridge ..." (Page 5, line 84) should be considered. The Glu5-Phe1 salt-bridge interaction was previously proposed in the Ng T4D model: perhaps state more clearly what the better resolution directly tells us.

4. In Figure 2, please show superpositions of the crystal structures of ΔN -Nm PilE/Ng PilE and ΔN -Nm PilE/Nm PilE MC58, so the structural differences reported in the text (page 6) can be visualized by readers. In addition, the numbering of the β -hairpins and β -strands should be stated in the text (page 6, lines 106-116).

5. Did the authors try flexible fitting of the models or only rigid body docking? What is the relationship of the refined cryo-EM subunits to the X-ray structure? How does this assembled model relate to the assembly proposed from antibodies? This would seem relevant for a human virulence factor, i.e. Assembly and antigenicity of the *Neisseria gonorrhoeae* pilus mapped with antibodies. Infect Immun. 1996 Feb;64(2):644-52.

Minor Comments:

1. In Figure 1A indicate where the hypervariable loop is in the D-region.
2. In Figure 3, show the surface representation color-coded by subunit for visualization.
3. What is the importance of the solvent channel in Nm T4P?
4. The manuscript is difficult reading for general readers. It should be re-written with more attention to keeping the biology in mind.

Reviewer #2 (Remarks to the Author):

This paper presents another high resolution X-ray crystal structure of a truncated pilin and a cryo-EM reconstruction of a pilus fibre. The cryo-EM reconstruction is at significantly higher resolution than those that already exist. Technically the paper is sound, but the biological impact is only moderate. As noted by the authors, there are at least three other crystal structures of highly similar pilins already. In the abstract three major claims of relevance are made ("this structure shows"): (1) that the PilE hypervariable region forms an exposed ridge on the pilus surface, (2) direct evidence of a salt bridge between two particular conserved residues, and (3) discovery of a melted portion of $\alpha 1$. Claim 1 is already known. Claim 2 is nice, but not very impactful, since we already understand salt bridges and protein packing well, and it is not clear why this salt bridge is particularly interesting. Claim 3 is most interesting, but I fully agree with the authors' own words (line 275) that it only "may help" to explain several

things - it remains unclear whether it ACTUALLY explains any of them. I don't understand the claim "The melting of this region likely explains.. electron transport" (line 296) since the model shows the key aromatic rings 8.6, 9.5, and 6.1 Angstroms away from each other, and then the text argues that distances less than 4.5 Angstroms are required for pi-pi orbital overlap. How is the melted region of the helix relevant to this? It was already clear that since the fibre can be extended by a factor of 3 without breaking, some structural change is happening. I agree the discovery of a melted region in alpha1 suggests that this is the weak point that would be the first to extend, but it is at most one of the changes needed, since the contacts between head groups would also have to change to allow the fibre to extend. In summary it is a nice structural advance, but the biological novelty and significance is only moderate.

Reviewer #3 (Remarks to the Author):

This is a logical and interesting extension of earlier work by several groups including the Egelman, Nassif and Craig labs to elucidate the molecular resolution structure of the assembled Type IV pilus. In this case the high-resolution structure of the soluble domain of *N. meningitidis* pilin subunit is solved by protein crystallography and the 6.5 Å resolution structure of the intact pilus from the same high adhesive strain is elucidated by electron microscopy (a feat unimaginable even a few years ago).

The literature is appropriately cited throughout and the story is clear (with a few figure exceptions, see below).

Overall the information is interesting because;

- 1) the stretch of the N-terminal alpha helix between residues 15 and 23 is modeled as unraveled in order to best match the electron microscopy data. Although kinks due to conserved Pro and Gly residues had been observed in all of the full length structures of pilin to date, some properties of the intact pilus can be explained by this unraveled stretch, which was not predicted.
- 2) The long-expected salt bridge between the N-terminal positive charge and the negative glutamate side chain at position 5 now is supported by some experimental evidence in the electron microscopy electron density.
- 3) Tyr 24 and 27 approach phe 1 from an adjacent monomer, making a helical band of low electron density. Is this possibly the molecular explanation for a low electron density at small radius described by Folkhard and Marvin to explain fiber diffraction data (JMB 1981).

On the other hand, some things are rigorous contributions to the canon but not novel, for example the strong similarity of the *N. meningitidis* pilin structure now presented to both the available *N. meningitidis* pilin structure from a different strain and the *N. gonorrhoeae* structure

Questions:

1. How are the-post translational modifications taken into account in the full pilus structure (e.g. glycosylations, methylation of N-terminus)? If they are not, why not?
2. The paragraph about the putative and relatively large conformational changes that might take place at pH2 in order to achieve an electrically conducting conformation (π - π stacking) of the two tyrosines 24 and 27 with phenylalanine 1 of an adjacent monomer are highly speculative and should be toned down.
3. The authors take the tack that the 6.5 Å resolution structure provides new and important information that could not be predicted based on the previous 12.5 Å structure from *N. gonorrhoeae*. I do not disagree with this but I think the converse is also true and important, that is to say these labs and others have done a very good job of predicting the molecular interactions and properties of the filament using other types of data before now, and the higher resolution structure does not contradict any earlier findings but rather extends them.
4. Accession numbers can't really be XXX and YYY. Please update.
5. Figure 2 legend typo? Should be Ser68 and not Ser69? If not, the figure is incorrectly labeled.
6. Figure 4 it is hard to follow which individual image goes with which legend part. Is the long central filament part of A or B or C?
7. Figure 6 has a legend for a part C but I don't see any Part C image.

Reviewers' comments and author's responses

Reviewer #1 (Remarks to the Author):

Neisseria meningitidis is a bacterial pathogen responsible for life-threatening diseases such as meningitis. The organism can cross the blood brain barrier (BBB) via the interaction of its Type IV pili (T4P) with the host endothelial cell receptor CD147 and the β 2 adrenergic receptor. PilE is the major component of T4P. The PilE protein is a ladle shaped molecule with a globular domain and a long N-terminal helix α 1. The mechanism of assembly of PilE and other pilin proteins to form the T4P, and the mode of interaction of T4P with host receptors is currently unknown. In the paper by Kolapan et al, the authors report the X-ray crystal structure of an N-terminally truncated form of the major pilin PilE protein from *Neisseria meningitidis* strain 8013 (Δ N-Nm PilE) at 1.44 Å resolution as well as a cryo-EM reconstruction of the intact T4P estimated at \sim 6.0 Å resolution. A model is derived from the cryo-EM map using the crystal structure of full length Ng PilE and their Δ N-Nm PilE structure. The crystal structure and cryo-EM model is compared to the previously published crystal structure of full length PilE from *Neisseria gonorrhoeae* (Ng PilE), and the previously published \sim 12.5 Å cryo-EM model of Ng T4P. \square Nm PilE has 78% overall sequence identity to Ng PilE and is 100% identical in the N-terminal helix α 1.

Key findings from the structural studies as presented by the authors are:

1. The crystal structure of the Δ N-Nm PilE is similar to the previously reported structure of Ng PilE. A 3.1 Å r.m.s.d is reported between the two structures for main chain atoms. The structural differences are limited to the β 3- β 4 loop that bends further away from a β -strand (which β -strand?) in Δ N-Nm PilE as compared to Ng PilE. and there are differences in a β -hairpin loop (which loop? - it was not specified). The value of the determined similarities and differences to understanding the biology should be clearly defined.
2. Their Δ N-Nm PilE structure is also similar (1.1 Å r.m.s.d for main-chain atoms) to the 1.43 Å Nm PilE structure from strain MC58. The two Nm PilE protein variants share 89% sequence identity. Minor conformational differences are noted in the text, but not described in any detail. Is there significance to the similarities or the minor differences?
3. A cryo-EM map was generated for the Nm T4P with an estimated resolution of \sim 6 Å. Rigid body fitting of the Ng PilE structure, followed by fitting in their Δ N-Nm PilE structure results in a model that shows the rod-like assembly of the T4P filament, as previously reported for Ng T4P. Well-defined density is observed for the PilE globular domain with ridge-like features, similar to those observed previously for Ng T4P. What is the new impact for the biology from this?
4. A notable difference between the Ng and Nm T4P is that the α -helical density appears to be weaker at the junction close to the globular domain and this is interpreted as loss of helical order and increased flexibility of α 1 in T4P.
5. A solvent channel that is approximately 11 Å in diameter is observed through the T4P filament.

6. The side chain of Glu5 in $\alpha 1$ is in close proximity to the Phe1 NH₂ and side-chain of Thr2 of $\alpha 1$ of a neighboring molecule, suggesting that a salt-bridge interaction may help stabilize the T4P assembly. This interaction was previously proposed to occur in the Ng T4P cryoEM model.

7. A stacking interaction is also observed between Phe1 of one subunit and Tyr24 and Tyr27 of the second subunit, suggesting that the π - π stacking also stabilizes the overall fold.

We did not mean to give the reviewer this impression – Phe1 on one subunit lies between the Tyr24/Tyr27 pair on two adjacent subunits, but not close enough for π - π stacking. We have clarified this point.

8. Residues 15-23 are reported to be flexible in the cryo-EM model of Nm T4P, a notable difference in the Nm PilE as compared to other PilE structures and models.

Despite the crystal structures available for PilE proteins and the cryo-EM model of the Ng T4P (at lower resolution), the paper has sufficient novelty for a high impact publication, but the biological significance needs to be clarified. The *Neisseria meningitidis* Type IV pilin and pilus structures together are of considerable biological importance. The Authors should consider the concerns/comments are listed below.

Major comments:

1. There is little biological insight to be gained from the structures as currently presented in the paper. For example, on page 9, the authors state that "This structure provides a molecular framework for understanding the coordinated action of PilE, PilV, and PilC..."; yet, it is not clear what new insight has been gained about the mechanism of assembly of T4P, that was not known from other studies.

We removed this statement as well as references to PilV and PilC. Certainly this structure will be important for understanding the coordinated action of PilE, V and C, but this understanding will also require more information – structural, localization and functional – regarding PilV and C.

Is there relevance to interpreting the movements defined from the related structures (Hexameric structures of the archaeal secretion ATPase GspE and implications for a universal secretion mechanism, EMBO J. 2007 Feb 7;26(3):878-90; Structure of an essential type IV pilus biogenesis protein provides insights into pilus and type II secretion systems. J Mol Biol. 2012 May 25;419(1-2):110-24). Structure of an essential type IV pilus biogenesis protein provides insights into pilus and type II secretion systems.). A model for assembly and disassembly of the pili should be presented based on the new and existing data, taking into account the observed flexibility of the helix $\alpha 1$.

We have included a figure (Fig. 6A) showing our updated assembly model, which takes into account the melting of the α -helix upon incorporation of the pilin subunit into the pilus filament. However, we have not (yet) derived insights from this structure regarding the assembly ATPase, which we maintain acts indirectly on the pilus filament via the polytopic inner membrane core protein (also known as the platform protein, PilG in Nm T4P).

How does this model relate to the assembly proposed from antibodies? This would seem relevant for a human virulence factor, i.e. Assembly and antigenicity of the *Neisseria gonorrhoeae* pilus mapped with antibodies. *Infect Immun.* 1996 Feb;64(2):644-52.

The N. meningitidis T4P structure is entirely consistent with the antibody mapping studies of N. gonorrhoeae T4P: antibodies to peptides 37-56, 94-108 bind to ends only, consistent with the locations of these epitopes on the top and bottom of the pilin domains, respectively, 110-120 bind to ends and sides (β 3- β 4 strands). Anti-pilus sera reacted to peptides 49-54 (exposed at the tip of the pilus), 72-77 (exposed α β -loop) and 121-145 (hypervariable β -hairpin, ridge). We discuss here in more detail the burial of the EYYLN epitope and a mechanism for the force-induced conformational change that occurs to reveal this epitope along the pilus length.

2. The authors should comment on whether they tried to crystallize full-length Nm PilE. What was the rationale for crystallization of the N-terminally truncated PilE protein? Is there structural information available from the related Nm PilE strain MC58 for the N-terminal portion of α 1 that would corroborate the flexibility observed in T4P for this region?

*We did not attempt to crystallize the full length Nm PilE so we did not mention it in the text. The 100% identity between Nm and Ng PilE in the N-terminal region, plus the availability of full length homologous Type IV pilins from *Pseudomonas aeruginosa* and *Dichelobacter nodosus* suggested to us that a full length Nm PilE structure would not provide enough new information to justify the effort required to purify sufficient amounts of *N. meningitidis* T4P, solubilize the pilin subunits and crystallize what is essentially a membrane protein. The MC58 PilE structure, like our PilE structure, lacks α 1N including residues 15-23 so provides no new information on the flexibility of this region.*

3. The authors discuss several specific side chain-side chain interactions such as salt-bridges and aromatic stacking interactions in the paper. At 6 Å resolution, such interactions may be postulated or inferred, but can they be stated to exist with certainty? For example, the words "direct evidence of a salt bridge ..." (Page 5, line 84) should be considered. The Glu5-Phe1 salt-bridge interaction was previously proposed in the Ng T4D model: perhaps state more clearly what the better resolution directly tells us.

We have modified this section to make it clear that this is an interpretation of the density and explaining that the rod-like density defining α 1N was not resolved in the 12.5 Å Ng T4P structure. We have also provided more information regarding the significance of the E5V interaction as per Reviewer 2.

4. In Figure 2, please show superpositions of the crystal structures of Δ N-Nm PilE/Ng PilE and Δ N-Nm PilE/Nm PilE MC58, so the structural differences reported in the text (page 6) can be visualized by readers. In addition, the numbering of the β -hairpins and β -strands should be stated in the text (page 6, lines 106-116).

The superposition has been added in a new Supplementary Fig. S1 and the numbering for the β -hairpins and more precisely described the small structural differences.

5. Did the authors try flexible fitting of the models or only rigid body docking? What is the

relationship of the refined cryo-EM subunits to the X-ray structure? How does this assembled model relate to the assembly proposed from antibodies? This would seem relevant for a human virulence factor, i.e. Assembly and antigenicity of the *Neisseria gonorrhoeae* pilus mapped with antibodies. *Infect Immun.* 1996 Feb;64(2):644-52.

*Rigid docking was followed by flexible fitting. Small adjustments were made in two loops of PilE to better fit the cryoEM density. These changes were described in the Methods and have now been added to the Results. Once the complete PilE model was built it was fit into the density as a rigid body. The PilE subunit in the refined Nm T4P model is very similar to the PilE crystal structure, as shown in the superposition in Fig. 4B. The only significant conformational change is a small movement in the $\beta 3$ - $\beta 4$ loop. The model is consistent with the antigenicity of *N. gonorrhoeae*, as described above.*

Minor Comments:

1. In Figure 1A indicate where the hypervariable loop is in the D-region.

Done.

2. In Figure 3, show the surface representation color-coded by subunit for visualization.

This cannot be readily done as the boundaries between subunits are not defined in the cryoEM map. The boundaries become defined by the atomic model, and this is shown in Figure 4 where subunits are colored individually.

3. What is the importance of the solvent channel in Nm T4P?

We suggest that it may provide room for the subunits to move in the core of the filament to allow for filament flexibility.

4. The manuscript is difficult reading for general readers. It should be re-written with more attention to keeping the biology in mind.

We have endeavored to do so in the revised version of the manuscript.

Reviewer #2 (Remarks to the Author):

This paper presents another high resolution X-ray crystal structure of a truncated pilin and a cryo-EM reconstruction of a pilus fibre.

We would like to point out that while there are a number of high resolution pilin structures there are in fact only two other T4P cryoEM reconstructions published, the next best being 12.5 Å resolution.

The cryo-EM reconstruction is at significantly higher resolution than those that already exist. Technically the paper is sound, but the biological impact is only moderate. As noted by the authors, there are at least three other crystal structures of highly similar pilins already. In the abstract three major claims of relevance are made ("this structure shows"): (1) that the PilE hypervariable region forms an exposed ridge on the pilus surface, (2) direct evidence of a salt

bridge between two particular conserved residues, and (3) discovery of a melted portion of $\alpha 1$. Claim 1 is already known.

*This was known for *N. gonorrhoeae* and data support it for *N. meningitidis* but it is shown directly for the first time here, including the 1.44 Å resolution structure of PilE in the context of the pilus filament. The hypervariable regions are not identical in the two pilins, and their differences are important in understanding Nm T4P interactions with the $\beta 2$ adrenergic receptor on endothelial cells. We have provided a more detailed description of this interaction based on the new structure.*

Claim 2 is nice, but not very impactful, since we already understand salt bridges and protein packing well, and it is not clear why this salt bridge is particularly interesting.

*We respectfully disagree with this reviewer's assessment. We thought that we understood the packing of the N-terminal α -helices well, but in fact we did not because until now the packing has never been visualized. It turns out to be significantly different from what we proposed previously for *N. gonorrhoeae* T4P based on the 12.5 Å resolution structure, in which the helices were not resolved. These differences have important implications for understanding pilus assembly and functions, which we have more clearly emphasized in the revised manuscript. The salt bridge has only been predicted, never demonstrated. This salt bridge is critical for efficient pilus assembly. Thus, providing direct evidence for this interaction is an important contribution to this field. In fact, a paper was published just this month on the role of Glu5 in the minor pilins of the Type II secretion system (Nivaskumar et al., Pseudopilin residue E5 is essential for recruitment by the type 2 secretion system assembly platform. Mol Micro 2016). We have endeavoured to clarify the significance of the salt bridge in our revision.*

Claim 3 is most interesting, but I fully agree with the authors' own words (line 275) that it only "may help" to explain several things - it remains unclear whether it ACTUALLY explains any of them.

*Time will tell whether it actually explains any of these. The new structure allows us to propose hypotheses that can be tested. For example, we can mutate residues that appear to be important for melting of $\alpha 1$ to see how these changes affect assembly and function. We can test our model for force-induced conformational change by examining force-induced exposure of other buried epitopes. The role of Glu5 in charge transfer can be investigated for *G. sulfurreducens* T4P. Knowing the precise structure of the $\beta 2$ adrenergic receptor binding site we can explore this interaction to understand how it triggers intracellular signalling.*

I don't understand the claim "The melting of this region likely explains.. electron transport" (line 296) since the model shows the key aromatic rings 8.6, 9.5, and 6.1 Angstroms away from each other, and then the text argues that distances less than 4.5 Angstroms are required for pi-pi orbital overlap.

*This section has been revised to improve clarity. Specifically, the distances between aromatic residues in the Nm T4P model are too great for pi-pi stacking, as they are for previously published T4P models. We suggest that the melted $\alpha 1$ segment will influence the position of Phe1 relative to the other aromatic residues and should thus be considered in future *G. sulfurreducens* T4P models. In addition, we reinforce the idea, brought forth in the recent *G. sulfurreducens* T4P models, that the symmetry of this filament must be quite different than that of the Ng and Nm*

T4P, with a much more compact structure, in order to achieve pi-pi stacking of the aromatic side chains.

How is the melted region of the helix relevant to this?

*Whether $\alpha 1:15-23$ is melted or helical will affect the position of Phe1 relative to the other aromatic residues. But more significantly, the model itself provides the most accurate data to date regarding the positions of the aromatic residues in the *N. meningitidis* T4P, which is very similar in sequence to the *G. sulfurreducens* T4P. This may serve as a better template to generate computational models for *G. sulfurreducens* T4P.*

It was already clear that since the fibre can be extended by a factor of 3 without breaking, some structural change is happening. I agree the discovery of a melted region in $\alpha 1$ suggests that this is the weak point that would be the first to extend, but it is at most one of the changes needed, since the contacts between head groups would also have to change to allow the fibre to extend. In summary it is a nice structural advance, but the biological novelty and significance is only moderate. It was not clear what structural change had to occur and in fact such a conformational change is difficult to imagine as it would require disruption of many subunit:subunit interactions.

We have provided more information on subunit:subunit interactions and a model for how force-induced conformational change might occur. The introduction of disorder in $\alpha 1$ results in a substantially different positioning of the most N-terminal part of this helix relative to our Ng T4P model, a positioning that is well-defined in the higher resolution structure. This changes the interactions among the $\alpha 1N$ and between $\alpha 1N$ and the globular domains. From the new model we are able to propose a force-induced conformational change that allows the pilus to extend, exposing the EYYLN epitope while maintaining critical subunit-subunit interactions.

Reviewer #3 (Remarks to the Author):

This is a logical and interesting extension of earlier work by several groups including the Egelman, Nassif and Craig labs to elucidate the molecular resolution structure of the assembled Type IV pilus. In this case the high -resolution structure of the soluble domain of *N. meningitidis* pilin subunit is solved by protein crystallography and the 6.5 Å resolution structure of the intact pilus from the same high adhesive stain is elucidated by electron microscopy (a feat unimaginable even a few years ago).

The literature is appropriately cited throughout and the story is clear (with a few figure exceptions, see below).

Overall the information is interesting because;

- 1) the stretch of the N-terminal α helix between residues 15 and 23 is modeled as unraveled in order to best match the electron microscopy data. Although kinks due to conserved Pro and Gly residues had been observed in all of the full length structures of pilin to date, some properties of the intact pilus can be explained by this unraveled stretch, which was not predicted.
- 2) The long-expected salt bridge between the N-terminal positive charge and the negative glutamate side chain at position 5 now is supported by some experimental evidence in the electron microscopy electron density.
- 3) Tyr 24 and 27 approach phe 1 from an adjacent monomer, making a helical band of low

electron density. Is this possibly the molecular explanation for a low electron density at small radius described by Folkhard and Marvin to explain fiber diffraction data (JMB 1981). Its not clear what the low electron density at ~ 31 Å is, as referred to in the Folkhard and Marvin paper. The aromatic side chains are distributed throughout the Nm pilus model. The “band of aromatic residues formed by Phe1, Tyr24 and Tyr27 is located between 18 and 30 Å. Interestingly, the disordered region of $\alpha 1$, residues 15-23, occupy a band at 22-36 Å.

On the other hand, some things are rigorous contributions to the canon but not novel, for example the strong similarity of the N. meningitidis pilin structure now presented to both the available N. meningitidis pilin structure from a different strain and the N. gonorrhoeae structure

Questions:

1. How are the-post translational modifications taken into account in the full pilus structure (e.g. glycosylations, methylation of N-terminus)? If they are not, why not?

We show the positions of Ser63 and Ser69 in the ΔN -PilE structure (Fig. 2A) and their positions are now also indicated in the cryoEM reconstruction Fig. 4A. We state in that legend that the resolution is insufficient to resolve these modifications.

2. The paragraph about the putative and relatively large conformational changes that might take place at pH2 in order to achieve an electrically conducting conformation (π - π stacking) of the two tyrosines 24 and 27 with phenylalanine 1 of an adjacent monomer are highly speculative and should be toned down.

Done.

3. The authors take the tack that the 6.5 Å resolution structure provides new and important information that could not be predicted based on the previous 12.5 Å structure from N. gonorrhoeae. I do not disagree with this but I think the converse is also true and important, that is to say these labs and others have done a very good job of predicting the molecular interactions and properties of the filament using other types of data before now, and the higher resolution structure does not contradict any earlier findings but rather extends them.

We thank the reviewer for this comment. We hope that the revised manuscript more clearly explains the novelty of this higher resolution structure and how it impacts our understanding of these critical bacterial appendages.

4. Accession numbers can't really be XXX and YYY. Please update.

The Nm PilE structure is 5JW8. The T4P reconstruction is an EMDB file, and this, along with the corresponding atomic filament model, will be deposited once submission is complete.

5. Figure 2 legend typo? Should be Ser68 and not Ser69? If not, the figure is incorrectly labeled.

Both the figure and the legend are correct. It is Ser69 in Nm PilE and Ser68 in Ng PilE.

6. Figure 4 it is hard to follow which individual image goes with which legend part. Is the long central filament part of A or B or C?

The figure has been modified to more clearly separate the panels belonging to A and B.

7. Figure 6 has a legend for a part C but I don't see any Part C image.

Fig. 6 has been replaced with models for T4P assembly and force-induced conformational change.

REVIEWERS' COMMENTS:

Reviewer #1 (Remarks to the Author):

In this revised manuscript, authors addresses most of the comments from the reviewers. And the written has become more clear and logical. However, authors should still consider the following comments before acceptance for publication.

1. Since the salt bridge interactions and N-terminal alpha 1 stretch between residues 15-23 are discussed and emphasized in this manuscript, the refinement Statistics of cryo-EM map of Nm PilE T4P with high resolution full-length Nm PilE model should be added.

2. How is the B-factor obtained from cryo-EM model refinement correlate with the crystal structure? The would be an useful parameter to understand the flexible region from the cryo-EM map. Maybe show it as B-factor distribution mapping (with color key) on the model.

3. Authors mentioned the melting of the helix at $\alpha 1:15-23$ appears to be induced upon integration of the pilin subunits into the pilus filament to facilitate packing of $\alpha 1N$ (Figure 6A) (line 249). How does the melt helix region link to proton motive force (electron transfer)? Would this account for more relax state or stress state? Maybe this could be incorporated in Figure 6 as well.

Reviewer #2 (Remarks to the Author):

The responses to reviewer critiques and revisions to the paper are all fine. The paper is sound and reports a moderate advance.

Reviewer #3 (Remarks to the Author):

The authors have largely addressed the reviewers' comments.

RESPONSE TO REVIEWERS

Reviewer #1 (Remarks to the Author):

In this revised manuscript, authors addresses most of the comments from the reviewers. And the written has become more clear and logical. However, authors should still consider the following comments before acceptance for publication.

1. Since the salt bridge interactions and N-terminal alpha 1 stretch between residues 15-23 are discussed and emphasized in this manuscript, the refinement Statistics of cryo-EM map of Nm PilE T4P with high resolution full-length Nm PilE model should be added.

These have been added in a new Table 2.

2. How is the B-factor obtained from cryo-EM model refinement correlate with the crystal structure? The would be an useful parameter to understand the flexible region from the cryo-EM map. Maybe show it as B-factor distribution mapping (with color key) on the model.

Although the resolution of the Nm T4P reconstruction, 6 Å, is substantially higher than any T4P structure reported previously it is nonetheless too low to refine B factors for the model fitting. We performed rigid body refinement with B factors fixed. This information has been added to the Methods section describing the model building.

3. Authors mentioned the melting of the helix at $\alpha 1:15-23$ appears to be induced upon integration of the pilin subunits into the pilus filament to facilitate packing of $\alpha 1N$ (Figure 6A) (line 249). How does the melt helix region link to proton motive force (electron transfer)? Would this account for more relax state or stress state? Maybe this could be incorporated in Figure 6 as well.

There is no evidence that proton motive force contributes to T4P assembly. Since there are no charged residues in $\alpha 1:15-23$ it seems unlikely that electron transfer would affect the conformation of this segment.

Reviewer #2 (Remarks to the Author):

The responses to reviewer critiques and revisions to the paper are all fine. The paper is sound and reports a moderate advance.

Reviewer #3 (Remarks to the Author):

The authors have largely addressed the reviewers' comments.